# Perinatal Adverse Effects in Newborns with Estimated Loss of Weight Percentile between the Third Trimester Ultrasound and Delivery. The GROWIN Study

**DOI:** 10.3390/jcm10204643

**Published:** 2021-10-10

**Authors:** María Sonsoles Galán Arévalo, Ignacio Mahillo-Fernández, Luis Mariano Esteban, Mercedes Andeyro-García, Roi Piñeiro Pérez, Miguel Saénz de Pipaón, Ricardo Savirón-Cornudella

**Affiliations:** 1Department of Neonatology, Villalba University General Hospital, 28400 Madrid, Spain; 2Biostatistics and Epidemiology Unit, Hospital Universitario Fundación, Jiménez Díaz and Fundación, Instituto de Investigación Sanitaria, 28040 Madrid, Spain; imahillo@fjd.es; 3Escuela Universitaria Politécnica, University of Zaragoza, La Almunia de Doña Godina, 50100 Zaragoza, Spain; lmeste@unizar.es; 4Department of Obstetrics and Gynecology, Villalba University General Hospital, 28400 Madrid, Spain; mercedes.andeyro@hgvillalba.es; 5Department of Paediatrics, Villalba University General Hospital, 28400 Madrid, Spain; roi.pineiro@quironsalud.es; 6Department of Neonatology, Hospital Universitario La Paz and Universidad Autónoma de Madrid, 28046 Madrid, Spain; miguel.saenz@salud.madrid.org; 7Department of Obstetrics and Gynecology, Hospital Clínico San Carlos and Instituto de Investigación Sanitaria San Carlos (IdISSC), Universidad Complutense, 28040 Madrid, Spain

**Keywords:** adverse perinatal outcomes, birthweight, estimated fetal weight, estimated percentile weight, fetal growth velocity, newborn, gestational age

## Abstract

Fetal growth restriction has been associated with an increased risk of adverse perinatal outcomes (APOs). We determined the importance of fetal growth detention (FGD) in late gestation for the occurrence of APOs in small-for-gestational-age (SGA) and appropriate-for-gestational-age (AGA) newborns. For this purpose, we analyzed a retrospective cohort study of 1067 singleton pregnancies. The newborns with higher APOs were SGA non-FGD and SGA FGD in 40.9% and 31.5% of cases, respectively, and we found an association between SGA non-FGD and any APO (OR 2.61; 95% CI: 1.35–4.99; *p* = 0.004). We did not find an increased APO risk in AGA FGD newborns (OR: 1.13, 95% CI: 0.80, 1.59; *p* = 0.483), except for cesarean delivery for non-reassuring fetal status (NRFS) with a decrease in percentile cutoff greater than 40 (RR: 2.41, 95% CI: 1.11–5.21) and 50 (RR: 2.93, 95% CI: 1.14–7.54). Conclusions: Newborns with the highest probability of APOs are SGA non-FGDs. AGA FGD newborns do not have a higher incidence of APOs than AGA non-FGDs, although with falls in percentile cutoff over 40, they have an increased risk of cesarean section due to NRFS. Further studies are warranted to detect these newborns who would benefit from close surveillance in late gestation and at delivery.

## 1. Introduction

Identifying newborns who have experienced intrauterine growth restriction is one of the greatest challenges in modern perinatology. The term fetal growth restriction (FGR) is used to define those newborns who have not reached their optimal intrauterine growth potential, and FGR can occur from the early to final stages of pregnancy [1,2,3] due to placental insufficiency [1,2,4,5,6]. It is estimated that 1 of every 10 pregnancies could present FGR and up to 20% in underdeveloped countries [7,8]. FGR is associated with increased rates of stillbirth, neonatal morbidity, and mortality [9,10,11,12]. In addition, individuals who have undergone FGR have poor long-term health outcomes, including risk of impaired neurodevelopment and a deterioration of cognitive abilities, a higher incidence of persistent short stature in childhood and adolescence, and cardiovascular and endocrine diseases in adulthood [13,14,15,16]. Correct identification of FGR can improve pregnancy surveillance [17,18] and the adequate treatment of associated complications, such as hypothermia or hypoglycemia, thereby improving long-term outcomes. [5].

The term small for gestational age (SGA) is usually used to refer to newborns whose birthweight is lower than the 10th centile [19,20]. Although most FGRs are SGA at delivery, some demographic factors, such as maternal age, parity, ethnicity, or body mass index, can be related to weight and height at birth [21,22], so SGA does not necessarily imply a pathological condition. Up to 70% are constitutionally small but healthy fetuses; that is, they have reached their full potential for intrauterine growth and are not at high risk of perinatal morbidity and mortality [23]. By applying the SGA definition, small but healthy newborns can be subjected to unnecessary interventions [5], as constitutionally small ones will be over diagnosed with SGA, and FGR will be underdiagnosed in fetuses with an estimated fetal weight (EFW) > 10th percentile [2].

Moreover, it has been postulated that some newborns with a weight percentile between 10 and 90 who have not reached their growth potential are actually FGR and are at increased risk of adverse outcomes [24]. Akolekar et al. observed that although the incidence of perinatal side effects is higher in SGA, most adverse perinatal outcomes occurred in AGAs [25].

Thus, fetal growth detention (FGD) has been considered as a reduction in EFW by 20 or more percentiles during the third trimester of pregnancy [26]. This cutoff point, or a higher one (30), has been taken in other studies, although with an interval of 6–8 weeks in the 3rd trimester, [26,27,28] or between the 2nd and 3rd trimester [29].

The present study determined whether newborns appropriate for gestational age (AGA) who have suffered FGD, defined as a reduction in estimated fetal growth by a decrease of 20 or more centiles between the 35th gestational week and delivery, had similar APOs to those diagnosed with SGA.

## 2. Materials and Methods

### 2.1. Study Design

The GROWth declINing Newborns (GROWIN) study was an observational, retrospective cohort study of births assisted at the Villalba University General Hospital between January 2015 and June 2017. Follow-up of the newborns included fetal growth up to delivery and their evolution in the first two years after delivery. The study was approved by the Clinical Research Ethics Committee of Fundación Jimenez Díaz (Madrid, Spain) (EO090-19_HGV) and the Central Research Commission of Primary Care Management (23/19).

The inclusion criteria were as follows: (i) live singleton pregnancies controlled in our center from the first trimester of gestation, (ii) fetal ultrasound assessment at gestational age of 35 (range 34–37) weeks; and (iii) deliveries between 37 and 42 weeks of gestational age with fetuses without structural malformations, chromosomal abnormalities, or metabolic diseases. This study followed the Strengthening the Reporting of Observational Studies in Epidemiology (STROBE) guidelines for cohort studies [30].

The variables collected in the study were maternal age, weight, height, and body mass index (BMI) at the beginning of pregnancy and parity, pregnancy and fetal pathology, type of delivery, cause of instrumental deliveries or cesarean sections. The last menstrual period was adjusted by the first trimester ultrasound [31] and an ultrasound screening was performed at 35 weeks (range 34–37 weeks) using either an ultrasound machine Voluson E8 (General Electric, Healthcare, Zipf, Austria) or a Toshiba Aplio500 (Toshiba Corporation, Tokio, Japan). EFW was calculated with the Hadlock et al. formula including biparietal diameter (BPD), head circumference (HC), abdominal circumference (AC), and femur length (FL) measurements [32]. The estimated percentile weight by ultrasound was calculated with a Spanish intrauterine growth standard [22] and the birthweight percentiles using a growth reference for the Spanish population [33]

We also collected perinatal outcomes to analyze APOs defined as the occurrence of 5 min Apgar score ≤ 7, arterial cord blood pH ≤ 7.10, instrumental or cesarean delivery for NRFS, basic and advanced neonatal resuscitation admission at birth in neonatal intensive care unit (NICU) due to asphyxia, sepsis, respiratory distress syndrome, and transient tachypnea. In addition, we collected newborn gender and somatometry at birth (birthweight, height, and cephalic circumference).

### 2.2. Study Groups

We considered the following patient groups based on percentile birthweight: (i) SGA newborns, those with a birthweight below the 10th percentile in the population of reference charts for that specific gestational age, with no fetal growth detention (SGA non-FGD); (ii) SGA newborns with fetal growth detention (SGA FGD) determined by a decrease of 20 or more centiles with respect to the estimated percentile weight in the third trimester ultrasound reflecting a fall in growth trajectory; (iii) appropriate-for-gestational-age (AGA), those with a birthweight between the 10th and 90th percentile, with no fetal growth detention (AGA non-FGD); (iv) AGA newborns with fetal growth detention (AGA FGD); and (v) large-for-gestational-age (LGA), defined as a birthweight above the 90th percentile.

### 2.3. Statistical Analyses

Data were extracted, and quantitative variables were described by medians and interquartile ranges (p25th–p75th), and qualitative variables were described by frequencies and percentages. Comparisons of quantitative variables were performed using the Mann–Whitney test when comparing two groups, or the Kruskal–Wallis test when comparing three or more groups. Qualitative variables were compared using the Chi-square test or the Fisher’s exact test. The predictive ability of AGA FGD newborns to diagnose APOs was analyzed using the area under receiver operating characteristics (ROC) curve and a threshold was sought for a specificity of 90%. A sensitivity (detection rate) of 10% was established for false positive rates (FPR). The percentile threshold point corresponding to the FPR values was also calculated.

The association between the five study groups and the presence of APOs was analyzed using univariate logistic regression models. In addition, we analyzed independent association using multivariate analysis adjusted by nulliparity, maternal BMI, hypertension, and diabetes. The models were summarized by the odds ratio, its 95% confidence interval and the *p*-value. The AGA non-FGD group was taken as a reference category for the odds ratio calculation.

Moreover, we explored how a greater decrease in percentile for the AGA FGD group could increase the risk ratio of APOs. We calculated the relative risk with a confidence interval at 95% for three decreases in percentile cutoffs (of 30, 40, and 50 centiles).

The significance level was set at 0.05 for all comparisons. Statistical analyses were performed using R v. 4.0.0 programming language (The R Foundation for Statistical Computing, Vienna, Austria).

## 3. Results

### 3.1. Descriptive Results

A total of 1067 pregnant women of 1528 eligible women were included in the present study (Figure 1).

The descriptive characteristics of the 1067 participants divided into the five study groups are shown in Table 1. There were no significant differences between the groups in terms of general characteristics, mother’s age, size, date of third trimester ultrasound scan, or type of delivery. However, significant differences were observed in mother’s weight and BMI at the beginning of pregnancy, which were lower in both SGA groups for weight and SGA non-FGD for BMI, in parity with a higher percentage of nulliparous in SGA groups (*p* < 0.001). Differences were also found in pregnancy pathology in terms of gestational diabetes, finding more cases in the LGA group (*p =* 0.014). SGA non-FGD had a median percentile at third trimester ultrasound (15.5) lower than SGA FGD had (44.5). In the study group AGA FGD, newborns were smaller than AGA non-FGD, with a median birthweight of 3105 (345) g vs. 3362 (465) g (*p* < 0.001) and median birthweight centile of 33.8 (27.1) vs. 60.4 (37.5) (*p* < 0.001), respectively.

### 3.2. Adverse Perinatal Outcomes

Table 2 describes the percentage of APOs in the five study groups and the differences between them. We found statistically significant differences between groups in relation to any adverse perinatal outcome (*p =* 0.050), instrumental (*p =* 0.030), cesarean deliveries for NRFS (*p =* 0.034), and admission at birth in NICU (*p =* 0.023). The newborns with higher APOs were SGA FGD (31.5%) and SGA non-FGD (40.9%).

Table 3 and Figure 2 display AUCs and sensitivity values plus the percentile threshold points for different FPR to predict total APOs in AGA FGD newborns. We found that AGA FGD had no predictive capacity for any APO (AUC 0.53; 95% CI, 0.49–0.57) with a decrease of at least 41.3 percentiles (90% specificity).

Figure 3 displays the results of the univariate logistic regression model and Figure 4 displays the results of the multivariate logistic regression model (adjusted by nulliparity, maternal BMI, hypertension, and diabetes) with the ORs, 95% CIs, and *p*-values to predict APOs. Evaluations in the multivariate analysis showed that SGA was the group with the highest risk for any APO, OR 1.49 (95% CI: 0.78, 2.75; *p* = 0.207) for SGA FGD and OR 2.61 (95% CI: 1.35, 4.99; *p* = 0.004) for SGA non-FGD, and only the result for SGA non-FGD was significant for any APO. We also found an increased risk in SGA non-FGD for instrumental delivery for NRFS (OR 4.67; 95% CI: 0.97,17.14; *p* = 0.030) or admission at birth in NICU (OR 5.46; 95% CI: 1.41, 17.8; *p* = 0.007). SGA FGD had a higher risk of cesarean delivery for NRFS (OR 2.99; 95% CI: 1.03, 7.72; *p* = 0.030) and advanced neonatal resuscitation (OR 3.76; 95% CI: 1.15, 10.7; *p* = 0.018). LGA presented a higher risk of basic neonatal resuscitation (OR 1.62; 95% CI: 1.00–2.60; *p* = 0.047).

We also estimated how a greater decrease in the percentile cutoff could increase the APO risk ratio for the AGA FGD group. Table 4 shows the relative risk of each amount of percentile decrease (30, 40, or 50 centiles) relative to each APO in the AGA FGD group. On the whole, we did not find differences for each individual APO regardless of the percentile decrease in AGA FGD newborns. The only exception was the cesarean delivery for NRFS, for which we observed a higher risk with decreases in percentile cutoffs of over 40 (RR: 2.41, 95% CI: 1.11–5.21) and 50 centiles (RR: 2.93, 95% CI: 1.14–7.54).

## 4. Discussion

### 4.1. Principal Findings

The newborns with the highest rate and risk of APOs were SGAs, and the only group that presented a significant risk corresponded to the SGA non-FGD group. We did not find an association between AGA FGD infants, determined by a decrease of 20 or more weight centiles between the third trimester ultrasound and delivery, and a higher incidence of APOs compared with AGA non-FGD newborns. However, AGA FGD newborns with decreases in percentile cutoffs of over 40 had an increased risk of cesarean section due to NRFS. Therefore, the risk of APO increases if the birthweight percentile is less than 10, but there is less evidence of APO risk related to the percentile fall between ultrasound at 35 weeks and delivery.

### 4.2. Birthweight Percentile and APOs

Our results regarding SGA newborns and other APOs, such as respiratory distress syndrome, arterial cord blood pH < 7.10, admission at birth, and prolonged stay in NICU were similar to those previously reported [8,34,35,36,37]

When SGA is suspected by ultrasound, these fetuses are closely monitored antenatally. By contrast, if AGA fetuses are suspected, there is not such a close follow-up. It has been suggested that newborns with an intrauterine restriction who do not reach their full genetic growth potential, in spite of a normal birthweight, are at higher risk of APOs than SGA [38,39]. We did not find increased APOs in AGA FGD compared with AGA non-FGD newborns, even with greater falls in percentile cutoff points, except for cesarean delivery for NRFS after falls of over 40 centiles. We could, therefore, consider birthweight to be a more important determinant of APO presentation than the percentile drop from 35 weeks to delivery per se.

### 4.3. Fetal Growth Percentile Deceleration and APOs

Prenatally undetected FGR is the most important risk factor for stillbirth. It increases the risk up to 8-fold and goes undiagnosed in most pregnancies, so preventive strategies should focus on improving its prenatal detection [12]. Fetal growth can be examined either by cross-sectionally measuring the fetal biometry [40] or by following the longitudinal growth of the fetuses with several measurements [41].

Fetal growth velocity is defined as the rate of fetal growth over a given time interval and is usually represented as a deviation from growth velocity charts (changes in centiles or Z-score with advancing gestation), which is significant to determine fetal growth [1,42].

In terms of fetal growth velocity and individual biometric measurements, there is an initial peak growth of BPD, HC, FL, and AC at 13, 14, 15, and 16 weeks, respectively, and BPD, HC, and AC have a second acceleration at 19 to 22, 19 to 21 and 27 to 31 weeks, respectively [43]. In general, normal growth in singletons increases from approximately 5 g/day at 14 to 15 weeks of gestation to 10 g/day at 20 weeks, peaking at 30 to 35 g/day at 32 to 34 weeks, after which the growth rates decrease [21].

According to the Delphi consensus [2], a fall of more than two quartiles in EFW in growth charts can diagnose these late onset FGR. Recently, the international Fetal Growth Velocity Increment Standards have been published from the Fetal Growth Longitudinal Study of the INTERGROWTH-21st project to facilitate the monitoring of fetal wellbeing exhaustively all over the world [42].

Although the cutoff point of 50 centiles is random, it shows the importance of growth velocity deceleration as a significant contributor to present APOs in SGA [35,44] and AGA fetuses [26,28,45]. Moreover, several previous studies have reported a link between fetal growth detection between ultrasounds and increased risk of APOs in both SGA and AGA newborns [36,37,44,46,47,48,49,50].

MacDonald et al. [28] concluded that a reduction in growth velocity between 28 and 36 weeks of pregnancy in AGA newborns is associated with increased risk of adverse outcomes, including stillbirth, so these fetuses may be undetected. When they investigated a decrease in EFW of less than 30 centiles, they found that it only occurs in 8.4% of the fetuses, suggesting it could be used as a tool with a good predictive value for detecting AGA in fetuses with placental insufficiency. We studied the decline in fetal growth from 35 weeks to birth in both SGA and AGA newborns with a cutoff of 20 centiles and did not find a clear association between APOs and this reduction. Hendrix et al. [51] noticed that abnormal fetal growth velocities, especially AC velocity between 20–32 weeks of gestational age, are associated with an adverse neonatal outcome of suspected AGA neonates resulting from suboptimal fetal growth. Our results are different because our observations were made later, from 35 weeks onwards, compared with those of Hendrix. Stratton et al. [26] observed that a fall of 20 or more centiles between two ultrasound scans in the 3rd trimester in AGA newborns was not associated with a higher risk of intrapartum complications, or perinatal morbidity, than those with adequate intrauterine growth, although there was an increased incidence of admission to NICU. We assessed the percentile drop from 35 weeks to delivery, but we did not study growth velocity. Our data suggest that late fetal growth detentions are associated with an increased risk of cesarean for NRFS in cases with a fall of more than 40 centiles. This could be explained by the fact that these are fetuses with some degree of placental insufficiency and are probably subjected to more stress during delivery than AGA non-FGD, which leads to an indication for cesarean section, but this insufficiency is presumably not as severe as in SGA. Probably, if the pregnancy were to continue, these fetuses would end up as SGA. However, they are not born with a percentile lower than 10 and this protects them from other APOs.

Chatzakis et al. [52] found that fetuses falling ≥ 50 estimated fetal weight centiles between the second and third trimester had increased rates of NICU admissions (OR 1.8) and perinatal death (OR 3.8), compared with AGA fetuses. This drop was a significant predictor of perinatal death and represents an increased risk of APOs. They investigated an earlier reduction in growth, in the phase of normal growth acceleration, which could justify different results to ours.

Pacora et al. [53] found that fetuses that suffered antenatal death at ≥20 weeks of gestation had decreased fetal growth velocity compared with controls. In addition, fetuses with an EFW percentile velocity < 10th centile had an increased risk of dying antepartum. They suggested that tools for better detection of fetuses at risk of death are needed. Verkauskiene et al. [29] performed an ultrasound scan every 4 weeks from week 22 to 36 and defined FGR as a drop of 20 centiles from 22 gestational weeks to birth. They concluded that FGR had an impact on body composition and hormonal parameters in AGA newborn infants, suggesting that these babies could have the same metabolic risk as SGA.

However, Bligh et al. [45], in a cohort of 436 women, demonstrated that decreased growth velocity in AGA fetuses at term (measured by EFW z-score change per week from 36 weeks until delivery) was associated with an increased risk of adverse neonatal outcomes. Although, unlike us, they studied the decreased z-score and compared the mean percentile loss between the APO and non-APO groups. We have not been able to demonstrate that AGA FGDs had the same APOs as SGAs, although there was a trend towards more adverse effects than AGA non-FGD, and in cases with a fall of more than 40 centiles, we found an increased risk of cesarean for NRFS. SGA non-FGD showed this low estimated weight from earlier stages of gestation than SGA FGDs, which had a later fall in weight, and, therefore, in cases where low birth weight was a consequence of placental insufficiency, this occurred at different times. For this reason, it is possible that SGA FGDs with a later weight loss may have less fetal impairment and fewer APOs. Thus, in the SGA group, the drop in percentile was also significant, showing that those with low percentiles from early gestation were the ones with the highest APOs.

Hence, clinical efforts should focus on the diagnosis and prediction of SGAs, especially those with the highest number of APOs, such as the SGA non-FGD group, the group of greatest interest, in order to proceed according to established protocols [54]. The best time to diagnose them would be the third trimester ultrasound. At this point, they are likely to have low estimated weight percentiles, although perhaps not all below the 10th percentile. Therefore, fetuses with low percentiles at 35 weeks ultrasound, even if they are not SGAs at that time, are worth following up. However, more studies are needed to determine the clinical implications of falls in percentile cutoffs of more than 40 points at the end of pregnancy in the AGA group.

Thus, in a specific group of pregnant women, it could be recommendable to repeat ultrasound prior to delivery. Furthermore, although they are a smaller group, identifying AGA newborns with an intrauterine growth restriction is important to be able to monitor them more closely in the third trimester. It is, therefore, important to continue to search for new methods of this monitoring, which would probably entail a combination of maternal, obstetric, and fetal factors.

### 4.4. Strengths and Limitations of the Study

Our research has several strengths as it includes a large sample size. We also defined the study groups following the longitudinal growth, not by cross-sectionally measuring the fetal biometry. We investigated the percentile decrease in both AGA and SGA newborns separately, on the basis of birth weight, so we do not have the bias of a percentile drop in newborns who are already SGAs and therefore have an increased risk of APOs based only on their percentile at birth.

Regarding limitations, this was a retrospective study, and when calculating EFW weight by ultrasound, an error of the technique can be assumed [55]. However, in similar studies that calculate the growth velocity between ultrasounds scans, this risk is taken on. Additionally, we had many patients in our sample with fetal growth detention, which could be due to interpersonal variability when performing ultrasound or an unrestrictive cutoff point (20 centiles). Moreover, we did not consider the time between ultrasound and delivery when calculating the percentile difference.

## 5. Conclusions

Newborns with the highest probability of APOs are SGAs. Of these, the only group that presents a significant risk are the SGA non-FGD newborns, which are therefore the most important group to detect antenatally. AGA FGD newborns do not have a higher incidence of APOs than AGA non-FGDs, although with decreases in percentile cutoffs of more than 40 points, they have an increased risk of cesarean section due to NRFS. Further research is required to determine whether this group would benefit from closer surveillance during late pregnancy and birth.

## Figures and Tables

**Figure 1 jcm-10-04643-f001:**
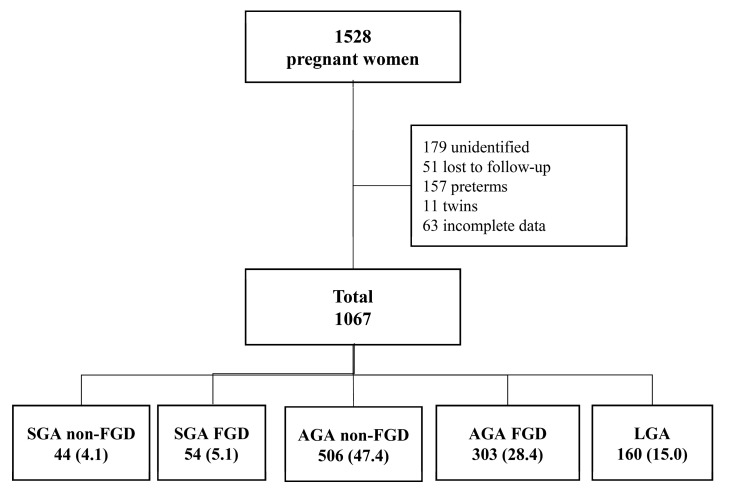
Flowchart of the study cohort. Data are reported as *n* (%). AGA FGD: appropriate for gestational age with fetal growth detention; AGA non-FGD: appropriate for gestational age without fetal growth detention; LGA: large for gestational age; SGA FGD: small for gestational age with fetal growth detention; SGA non-FGD: small for gestational age without fetal growth detention.

**Figure 2 jcm-10-04643-f002:**
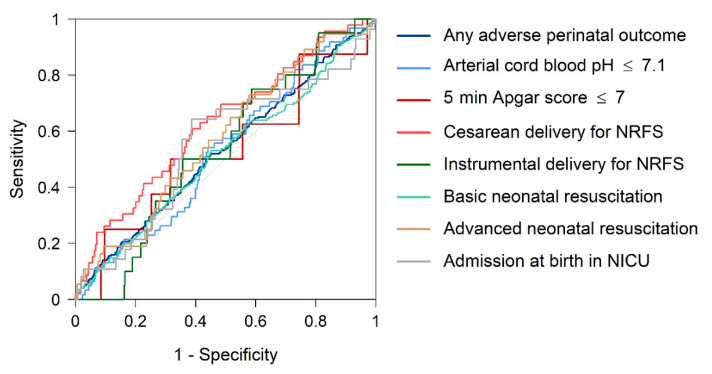
Receiver operating characteristic curves: fetal growth detention as predictor of adverse perinatal outcomes (APOs). NRFS: non-reassuring fetal status.

**Figure 3 jcm-10-04643-f003:**
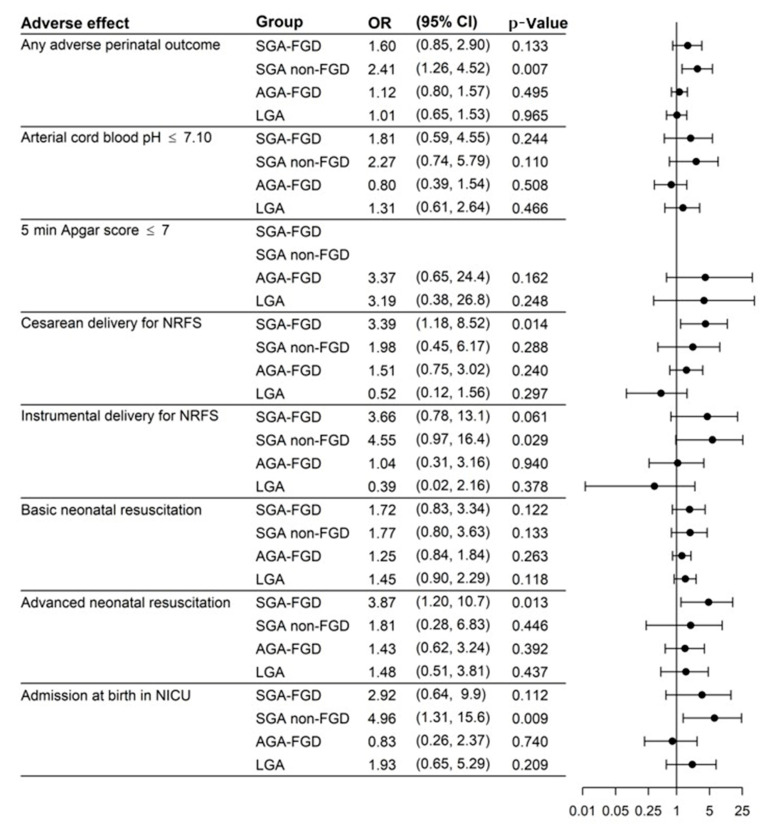
Results of the logistic regression model (univariate analysis). AGA FGD: appropriate for gestational age with fetal growth detention; AGA non-FGD: appropriate for gestational age without fetal growth detention; LGA: large for gestational age; NICU: neonatal intensive care unit; NRFS: non-reassuring fetal status; SGA FGD: small for gestational age with fetal growth detention; SGA non-FGD: small for gestational age without fetal growth detention.

**Figure 4 jcm-10-04643-f004:**
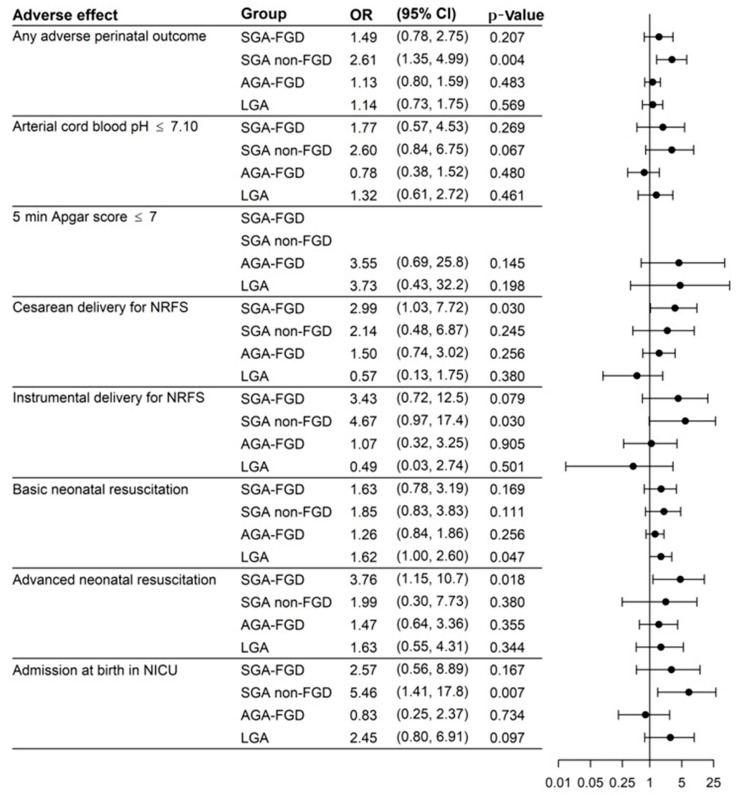
Results of the logistic regression model (multivariate analysis adjusted by nulliparity, maternal BMI, hypertension, and diabetes). AGA FGD: appropriate for gestational age with fetal growth detention; AGA non-FGD: appropriate for gestational age without fetal growth detention; LGA: large for gestational age; NICU: neonatal intensive care unit; NRFS: non-reassuring fetal status; SGA FGD: small for gestational age with fetal growth detention; SGA non-FGD: small for gestational age without fetal growth detention.

**Table 1 jcm-10-04643-t001:** Descriptive baseline characteristics. Data are reported as *n* (%) or median and interquartile range (p25th–p75th). AGA FGD: appropriate for gestational age with fetal growth detention; AGA non-FGD: appropriate for gestational age without fetal growth detention; LGA: large for gestational age; NRFS: non-reassuring fetal status; SGA FGD: small for gestational age with fetal growth detention; SGA non-FGD: small for gestational age without fetal growth detention.

Variable	Total	SGA-FGD	SGA Non-FGD	AGA FGD	AGA Non-FGD	LGA	*p*-Value
Cases	1067	54 (5.1)	44 (4.1)	303 (28.4)	506 (47.4)	160 (15.0)	
Maternal characteristics							
Parity							<0.001
Nulliparous	626 (58.7)	41 (75.9)	29 (65.9)	181 (59.7)	307 (60.7)	68 (42.5)	
Multiparous	441 (41.3)	13 (24.1)	15 (34.1)	122 (40.3)	199 (39.3)	92 (57.5)	
Maternal weight (kg)	66.4 (15.0)	62.6 (13.8)	60.0 (13.7)	65.4 (14.1)	66.2 (15.0)	68.0 (21.0)	<0.001
Height (cm)	163 (3.00)	163 (5.50)	163 (2.00)	163 (3.00)	163 (2.00)	163 (4.25)	0.528
Body mass index	25.0 (5.68)	24.5 (5.08)	22.2 (5.61)	24.5 (5.05)	25.2 (5.51)	25.6 (6.75)	<0.001
Age at delivery (years)	32.5 (8.11)	34.8 (8.20)	32.2 (5.38)	33.5 (8.80)	32.2 (7.94)	32.0 (6.10)	0.075
Pregnancy pathology							
Gestational diabetes	89 (8.3)	2 (3.7)	4 (9.1)	16 (5.3)	44 (8.7)	23 (14.4)	0.014
Hypertensive disorders of pregnancy	43 (4.0)	4 (7.4)	1 (2.3)	15 (5.0)	13 (2.6)	10 (6.2)	0.082
Type of delivery							0.346
Normal vaginal delivery	702 (65.8)	28 (51.9)	27 (61.4)	199 (65.7)	342 (67.6)	106 (66.2)	
Instrumental delivery	224 (21.0)	13 (24.1)	11 (25.0)	63 (20.8)	106 (20.9)	31 (19.4)	
Cesarean section	141 (13.2)	13 (24.1)	6 (13.6)	41 (13.5)	58 (11.5)	23 (14.4)	
Ultrasound parameters at 35 weeks							
Gestational age (days)	246 (5.00)	246 (3.00)	247 (6.25)	246 (4.00)	246 (5.00)	246 (5.25)	0.525
Estimated fetal weight (g)	2571 (333)	2421 (138)	2200 (268)	2625 (240)	2513 (321)	2776 (404)	<0.001
Estimated fetal weight percentile	65.0 (40.9)	44.5 (27.7)	15.5 (12.0)	72.9 (24.5)	58.3 (38.6)	88.5 (20.2)	<0.001
Delivery data							
Gestational age at birth (days)	278 (12.0)	280 (13.0)	273 (15.0)	278 (11.0)	278 (12.0)	276 (12.0)	0.008
Birthweight (g)	3285 (595)	2712 (194)	2518 (251)	3105 (345)	3362 (465)	3935 (355)	<0.001
Birthweight percentile	53.2 (54.4)	6.04 (3.57)	4.55 (5.56)	33.8 (27.1)	60.4 (37.5)	95.4 (5.84)	<0.001
Arterial cord blood pH	7.27 (0.11)	7.26 (0.12)	7.25 (0.10)	7.27 (0.10)	7.27 (0.11)	7.26 (0.12)	0.133
5 min Apgar score ≤ 7	10.0 (0.00)	10.0 (0.00)	10.0 (0.25)	10.0 (0.00)	10.0 (0.00)	10.0 (0.00)	0.069
Newborn Sex							0.001
Male	577 (54.1)	26 (48.1)	22 (50.0)	141 (46.5)	307 (60.7)	81 (50.6)	
Female	490 (45.9)	28 (51.9)	22 (50.0)	162 (53.5)	199 (39.3)	79 (49.4)	
Newborn length (cm)	49.0 (2.00)	47.0 (2.00)	47.0 (1.62)	49.0 (2.50)	49.0 (2.00)	50.5 (2.12)	<0.001
Newborn head circumference (cm)	34.0 (1.50)	33.5 (1.38)	33.0 (1.78)	34.0 (2.00)	34.0 (1.38)	35.0 (2.00)	<0.001

**Table 2 jcm-10-04643-t002:** Percentage of adverse perinatal outcomes (APOs) in the study groups and the differences between them. Data are reported as *n* (%). AGA FGD: appropriate for gestational age with fetal growth detention; AGA non-FGD: appropriate for gestational age without fetal growth detention; LGA: large for gestational age; NICU: neonatal intensive care unit; NRFS: non-reassuring fetal status; SGA FGD: small for gestational age with fetal growth detention; SGA non-FGD: small for gestational age without fetal growth detention.

Adverse Effects	Total	SGA FGD	SGA Non-FGD	AGA FGD	AGA Non-FGD	LGA	*p*-Value
Any adverse perinatal outcome	258 (24.2)	17 (31.5)	18 (40.9)	74 (24.4)	113 (22.3)	36 (22.5)	0.050
Arterial cord blood pH ≤ 7.10	61 (5.7)	5 (9.3)	5 (11.4)	13 (4.3)	27 (5.3)	11 (6.9)	0.236
5 min Apgar score ≤ 7	8 (0.7)	0 (0.0)	0 (0.0)	4 (1.3)	2 (0.4)	2 (1.2)	0.475
Cesarean delivery for NRFS	46 (4.3)	6 (11.1)	3 (6.8)	16 (5.3)	18 (3.6)	3 (1.9)	0.034
Instrumental delivery for NRFS	20 (1.9)	3 (5.6)	3 (6.8)	5 (1.7)	8 (1.6)	1 (0.6)	0.030
Basic neonatal resuscitation	177 (16.6)	12 (22.2)	10 (22.7)	52 (17.2)	72 (14.2)	31 (19.4)	0.245
Advanced neonatal resuscitation	37 (3.5)	5 (9.3)	2 (4.5)	11 (3.6)	13 (2.6)	6 (3.)	0.144
Admission at birth in NICU	28 (2.6)	3 (5.6)	4 (9.1)	5 (1.7)	10 (2.0)	6 (3.8)	0.023

**Table 3 jcm-10-04643-t003:** Values of area under the receiver operating characteristic curves (AUCs) and sensitivities plus the percentile threshold point for different false positive rates (FPR) to predict total adverse perinatal outcomes (APOs) by fetal growth detention. NICU: neonatal intensive care unit; NRFS: non-reassuring fetal status.

Adverse Effect	AUC (95% CI)	Sensitivity a FPR 10% (Specificity 90%)	Birthweight Percentile Threshold
Any adverse perinatal outcome	0.53 (0.49, 0.57)	14%	41.3
Arterial cord blood pH ≤ 7.10	0.53 (0.46, 0.60)	13%	42.0
5 min Apgar score ≤ 7	0.53 (0.30, 0.76)	25%	42.0
Cesarean delivery for NRFS	0.63 (0.54, 0.71)	26%	41.4
Instrumental delivery for NRFS	0.54 (0.43, 0.65)	0%	42.4
Basic neonatal resuscitation	0.52 (0.47, 0.57)	13%	41.8
Advanced neonatal resuscitation	0.57 (0.48, 0.66)	19%	41.8
Admission at birth in NICU	0.55 (0.44, 0.67)	7%	47.1

**Table 4 jcm-10-04643-t004:** Relative risk of each amount of percentile decreases relative to each APO in the AGA FGD group.

	Percentile Decrease Cutoff		
Adverse Effect	30	40	50
Any adverse perinatal outcome	1.18 (0.88, 1.58)	1.36 (0.98, 1.90)	1.40 (0.89, 2.19)
Arterial cord blood pH≤7.10	1.24 (0.64, 2.39)	1.25 (0.56, 2.79)	1.17 (0.37, 3.72)
5 min Apgar score ≤ 7	2.78 (0.39, 19.59)	4.82 (0.69, 33.8)	
Cesarean delivery for NRFS	1.70 (0.82, 3.53)	2.41 (1.11, 5.21)	2.93 (1.14, 7.54)
Instrumental delivery for NRFS	0.70 (0.15, 3.24)		
Basic neonatal resuscitation	1.24 (0.84, 1.81)	1.41 (0.91, 2.18)	1.32 (0.70, 2.47)
Advanced neonatal resuscitation	1.07 (0.39, 2.96)	1.85 (0.68, 5.09)	1.62 (0.38, 6.98)
Admission at birth in NICU	0.83 (0.23, 3.00)	0.96 (0.21, 4.33)	

## Data Availability

The data analyzed were retrieved from the Villalba University General Hospital database.

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
