# Peer review of "Perinatal Adverse Effects in Newborns with Estimated Loss of Weight Percentile between the Third Trimester Ultrasound and Delivery. The GROWIN Study"

_jcm, 2021, doi:10.3390/jcm10204643_

Round 1

Reviewer 1 Report

Whether SGA is diagnosed with FGR during pregnancy is an important problem. The prognosis of newborns was reported at JAMA in 2021, and it is a field that is drawing attention. I agree that surveillance in late pregnancy is important. Minor ・ You need to explain why FGR is suspected too high for the total number of deliveries. ・ It is a well-known fact that APO increases in SGA and severe SGA. I think the increase in Caesarean section with AGA FGD is new findings, but ... ・ You should analyze SGA separately for SGA non FGD and SGA FDG. ・ You need to explain carefully why cesarean section increases in AGA FGD.

Reviewer 2 Report

The Authors presents results of a retrospective observational study addressing problem highly relevant for perinatal care.

The manuscript is carefully written and the Authors attempted a comprehensive statistical analysis. However, there are several issues which need to be addressed before a final decision regarding the manuscript could be made:

1/ the Authors performed a wide range of statistical tests to test for any association between FGD and APO. However, there is no information regarding the logistic regression whether the Authors adjusted the models for maternal characteristics which significantly differed across the groups (nulliparity, maternal BMI). If this has not been done, performing this analysis could strongly enhance scientific value of the manuscript.

2/Did the Authors also considered any chronic maternal diseases like hypertension or diabetes in their analysis, or were these women excluded from the analysis? If not, then again - controlling for these factors could make the statistical analysis stronger. 

3/ Did the Authors attempted any power calculation to justify their study sample size?

4/ There is an interesting shift in neonatal sex between non-FGD AGA and FGD-AGA newborns. What could be a reason for this finding?

5/ If the descriptive statistics presented in the Table 2 were compared across the five groups, then p=0.046, p=0.030, p=0.027 can no longer be considered significant.

6/ The Authors provide clear and carefully designed visual presentations of their results. However, the Figure 2 does not add anything to the manuscript and can be omitted.

Round 2

Reviewer 1 Report

The manuscript has been revised well. I think this manuscript will be acceptable.

Reviewer 2 Report

Dear Authors,

thank you for your detailed response. I consider my comments fully addressed.